# Effect on Broiler Production Performance and Meat Quality of Feeding *Ulva lactuca* Supplemented with Carbohydrases

**DOI:** 10.3390/ani12131720

**Published:** 2022-07-02

**Authors:** Mónica M. Costa, José M. Pestana, Patrícia Carvalho, Cristina M. Alfaia, Cátia F. Martins, Daniela Carvalho, Miguel Mourato, Sandra Gueifão, Inês Delgado, Inês Coelho, José P. C. Lemos, Madalena M. Lordelo, José A. M. Prates

**Affiliations:** 1CIISA—Centro de Investigação Interdisciplinar em Sanidade Animal, Faculdade de Medicina Veterinária, Universidade de Lisboa, 1300-477 Lisboa, Portugal; monicacosta@fmv.ulisboa.pt (M.M.C.); jpestana@fmv.ulisboa.pt (J.M.P.); cpmateus@fmv.ulisboa.pt (C.M.A.); catiamartins@isa.ulisboa.pt (C.F.M.); jpclemos@fmv.ulisboa.pt (J.P.C.L.); 2Laboratório Associado para Ciência Animal e Veterinária (AL4AnimalS), Faculdade de Medicina Veterinária, Universidade de Lisboa, 1300-477 Lisboa, Portugal; 3LEAF—Linking Landscape, Environment, Agriculture and Food, Instituto Superior de Agronomia, Universidade de Lisboa, 1349-017 Lisboa, Portugal; patysofy@gmail.com (P.C.); danielacarvalho@isa.ulisboa.pt (D.C.); mmourato@isa.ulisboa.pt (M.M.); mlordelo@isa.ulisboa.pt (M.M.L.); 4INSA—Departamento de Alimentação e Nutrição, Instituto Nacional de Saúde Doutor Ricardo Jorge, Avenida Padre Cruz, 1649-016 Lisboa, Portugal; sandra.gueifao@insa.min-saude.pt (S.G.); ines.delgado@insa.min-saude.pt (I.D.); ines.coelho@insa.min-saude.pt (I.C.)

**Keywords:** *Ulva lactuca*, carbohydrase, poultry growth, meat quality, broiler

## Abstract

**Simple Summary:**

Macroalgae have been increasingly exploited worldwide for feed, food and biofuel applications, due to their nutritive and bioactive compounds. Green seaweeds belonging to the genus *Ulva* have high growth rates, which makes them suitable for being cultured in sustainable algae production, such as an integrated multi-trophic aquaculture system. This is expected to increase the use of *Ulva* sp. as an alternative source to conventional feedstuffs (e.g., cereals and soybean meal) for poultry. The objective of the current study was to assess if the incorporation of 15% *Ulva lactuca* in broiler chickens’ diet, combined or not with carbohydrate-active enzymes, would enhance meat nutritional quality without compromising animal growth performance. Overall, *U. lactuca* led to an accumulation of antioxidant carotenoids, n-3 PUFA and macrominerals, including magnesium, potassium and phosphorus, in the breast muscle, with likely health benefits, without significantly impairing growth performance. The supplementation of macroalgae with a recombinant ulvan lyase reduced ileal viscosity with possible beneficial effects on broiler digestibility. Although dietary *U. lactuca* showed potential to increase meat quality, it reduced meat overall acceptability, which suggests the use of a lower algae inclusion level to prevent a negative meat sensory perception for consumers.

**Abstract:**

The aim of the study was to test if feeding 15% *U. lactuca* to broilers, alone or combined with carbohydrases, enhanced meat nutritional quality, without compromising growth performance. One hundred and twenty 22-day-old broilers were allocated to the following diets and replicated 10 times for 14 days: (1) maize and soy-based diet (control); (2) control with 15% *U. lactuca* (UL); (3) UL diet with 0.005% commercial carbohydrase mixture (ULC); and (4) UL diet with 0.01% ulvan lyase (ULE). Final body weight and average daily gain decreased (*p* < 0.050) with the ULE diet compared with the control, but no significant differences were found for the other diets. The intestinal viscosity increased (*p* < 0.001) with all alga diets but was lowered (*p* < 0.050) in the ileum with the ULE diet, relative to UL and ULC diets. Meat lightness and redness values, off-flavours, and total carotenoids increased (*p* < 0.001), while yellow values, tenderness, juiciness, overall acceptability, α- and γ-tocopherol, and total lipids decreased (*p* < 0.001) with alga diets. The n-3 polyunsaturated fatty acids (PUFA) increased (*p* < 0.050), and the n-6/n-3 PUFA ratio decreased (*p* < 0.001) with the ULE diet. Total minerals in meat increased (*p* < 0.001) with alga diets, conversely to sodium and zinc (*p* < 0.001). Feeding 15% of *U. lactuca* to broilers did not impair growth but increased meat nutritional value through the accumulation of health-promoting antioxidant carotenoids, n-3 PUFA and total minerals, although reducing overall meat acceptability.

## 1. Introduction

Seaweeds have been exploited for several purposes, including feed, food and biofuel applications, due to their nutritive and bioactive compounds [1]. In particular, green macroalgae, mainly the genus *Ulva*, have high growth rates leading to rapid biomass accumulation [2], which caused them to be increasingly produced worldwide [3]. Although there are environmental and economic impacts of seaweed production, the use of integrated multi-trophic aquaculture (IMTA) offers a more sustainable production system for *Ulva* sp. [4]. The nutritional composition of *Ulva* sp. is variable but, in general, these algae are good sources of chlorophylls *a* and *b*, carotenoids and minerals, such as iodine [1]. Although the protein content is, on average, 15.8% dry matter (DM), for some species, such as *U. lactuca* and *U. prolifera*, crude protein can reach values of 31.6 to 41.8% DM [5]. Total lipids are found in low amounts (up to 6.6% DM) in green seaweeds but with a healthy profile of beneficial polyunsaturated fatty acids (PUFA) [6]. The bioactive constituents of *Ulva* sp. are mainly antioxidant pigments [7] and the sulphated polysaccharide ulvan. This polymer has many human health-promoting functions, including immunomodulation and antioxidant activity [8].

Considering the benefits of using *Ulva* sp. as a feed ingredient, several studies reported their impact on poultry growth or meat quality [9,10,11,12,13,14,15]. However, only a few assessed the effect of incorporating over 10% of *Ulva* sp. in feeds [9,10]. Ventura et al. [10] observed no negative effect on broilers’ growth of feeding 10% of *Ulva rigida*, but higher algae levels compromised animal growth. In fact, high inclusion levels of green seaweeds may impair nutrient digestibility as a result of indigestible algae cell wall polysaccharides [16], which mainly comprise gel-forming ulvan and insoluble cellulose, together with low amounts of xyloglucan and glucuronans [17]. Therefore, the use of Carbohydrate-Active enZymes (CAZymes) emerges as a possible solution to degrade the *Ulva* sp. cell wall, due to their efficiency in hydrolysing *Ulva* sp. biomass for protein and carbohydrate extractions [18,19,20]. In addition, the in vitro ability of a single ulvan lyase from the family of 25 polysaccharide lyases (PL25) to partially disrupt the *U. lactuca* cell wall and release mono- and oligosaccharides, and monounsaturated fatty acids (e.g., C18:1*c*9), was recently reported [21]. Furthermore, CAZymes have shown activity towards carbohydrate present in microalgae cell walls [22,23] and in grains [24,25]. Thus, the degradation of seaweed biomass with feed enzymes would optimize their utilization as feedstuffs to partially replace unsustainable and conventional sources such as maize and soybean meal [1]. However, to the best of our knowledge, the effect of feeding macroalgae supplemented with CAZymes on broiler performance and meat quality was only evaluated in a few studies using *Ulva* sp. [26,27] or a brown seaweed, *Laminaria digitata* [28].

In the present study, we hypothesized that the combination of a previously in vitro tested ulvan lyase (PL25) [21], or a commercial carbohydrase mixture with dietary *U. lactuca* replacing 15% of maize and soybean meal, increases the nutritive value of seaweed for broilers with a consequent improvement of meat quality through the deposition of health promoting bioactive compounds, without compromising animal growth performance.

## 2. Materials and Methods

### 2.1. Animals and Experimental Diets

The procedures with animals were authorized by both the ISA Ethics Commission and ORBEA (protocol code number 1/ORBEA-ISA/2020, date of approval 7 July 2020) and the Portuguese National Authority for Animal Health (DGAV), following the 2010/63/EU Directive.

The experiment was conducted as described in previous reports [29,30]. Briefly, one hundred and twenty 1-day-old male Ross 308 broiler chicks were weighed (initial body weight, BW (d 0) of 39.8 ± 0.20 g), individually tagged, and 3 broilers were housed per pen in a total of 40 battery cages for 35 days, under a controlled environment. The number of animals used in the experiment was chosen to follow the 3R´s principle and in accordance with previous reports [29,30]. After 21 days of animal feeding with an adapted maize and soy-based diet, and during the finishing period, broilers were fed *ad libitum* one of 4 dietary treatments (*n* = 10) for 14 days: (1) maize and soy-based diet (control); (2) control with 15% *U. lactuca* powder (Algolesko; Plobannalec-Lesconil, Brittany, France) (UL); (3) UL diet added with 0.005% commercial CAZyme mixture (ULC); and (4) UL diet added with 0.01% ulvan lyase (ULE). The commercial carbohydrase mixture (Rovabio^®^, Adisseo; Antony, France) contained, as main active enzymes, endo-1,3(4)-β-glucanase (EC 3.2.1.6) (4300 U/g) and endo-1,4-β-xylanase (EC 3.2.1.8) (3200 U/g), whereas ulvan lyase is a recombinant CAZyme (PL25 family) capable of releasing reducing sugars (4.54 g/L) and mono- and oligosaccharides (22.6 mmol /100 g dried alga) from *U. lactuca* biomass [21]. The mash experimental diets were formulated to be isocaloric and isonitrogenous and their ingredients are shown in Table 1.

Growth performance parameters including average daily gain (ADG), average daily feed intake (ADFI) and feed conversion ratio (FCR) were determined by weighing the animals and feeders weekly. After the experimental period, 1 animal per pen was stunned and slaughtered by exsanguination. Gastrointestinal organs were manually excised and rinsed with tap water to obtain their weight and length. The digesta viscosity of duodenum plus jejunum, and ileum was evaluated with a viscometer (Brookfield Engineering Laboratories, Middleboro, MA, USA.), as described in previous reports [29]. Carcasses were cooled to an internal temperature of 4 °C, in an air-chilled circuit, which was controlled using a probe thermometer. The breast muscle (*pectoralis major*) was dissected from the right side of carcasses, for carcass traits and sensory analyses, and from the left side, for determination of meat composition and lipid oxidation. This muscle was selected because of its high carcass representativeness (weight percentage), which makes it the most consumed muscle in proportion to total carcass.

### 2.2. Production of Recombinant Ulvan Lyase

*Escherichia coli* (BL21) cells, previously transformed with plasmids containing the recombinant ulvan lyase gene, were grown on Luria–Bertani (LB) medium until mid-exponential phase, as recently reported [28]. Expression of the recombinant gene was auto induced in an LB medium (Nzytech, Lisbon, Portugal) with overnight incubation at 19 °C, 140 rpm. Cells were disrupted by ultrasonication, and then centrifuged to obtain the supernatant with the protein extract. The extract was freeze-dried and incorporated in the ULE diet at 0.01% feed.

### 2.3. Chemical Composition of U. lactuca and Diets

The proximate analysis of seaweed and diets is shown in Table 2. Macroalgae and diets were analysed for their DM, crude protein, crude fat, ash and gross energy using AOAC procedures [31]. The metabolizable energy was determined using a previously described equation [28]. Amino acid profile in the diet was expressed as estimated available percentages. Fatty acids in *U. lactuca* and diets were extracted and trans-esterified by one-step acidic methylation. The nonadecanoic acid (19:0) methyl ester was used as the internal standard. Then, fatty acid methyl esters (FAME) were injected into a capillary column (Supelco, Bellefonte, PA, USA.) incorporated in a gas chromatograph with flame ionization detector (GC-FID), following the previously specified chromatographic conditions [30]. The results were expressed as percentage of total fatty acids.

β-Carotene and vitamin of macroalgae and diets were extracted in duplicate from 100 mg of samples, as described by Prates et al. [32]. Briefly, ascorbic acid and a saponification solution were added to the samples followed by incubation at 80 °C, 15 min, and centrifugation for the separation of n-hexane phase. Then, the organic phase was filtered and injected in a Zorbax RX-Sil column (Agilent Technologies Inc., Palo Alto, CA, USA.) with a set of 2 serial detectors incorporated in an HPLC system, according to previously reported conditions [29]. A standard curve of peak area vs. concentration based on an external standard technique was used for the determination of the amounts of β-carotene and diterpenes.

Pigments of *U. lactuca* and diets were extracted with acetone using a procedure described by Pestana et al. [29]. Succinctly, 0.5 g of sample was agitated overnight with the extraction solvent and centrifuged for separation of the supernatant. Detection of chlorophyll a and b and total carotenoids was performed with a spectrophotometer (Ultrospec 3100 pro, Amersham Biosciences, Little Chalfont, UK), at 662, 645 and 470 nm, respectively. Pigment concentrations were determined using the formulas reported by Dere et al. [33].

Macro- and micro- minerals of *U. lactuca* and diets were analysed in accordance with Ribeiro et al. [34], and, for iodine and bromine, with Delgado et al. [35]. Succinctly, all the elements, except iodine and bromine, were extracted, in triplicate, from 0.3 g of sample digested with concentrated nitric and hydrochloric acids, and with hydrogen peroxide added afterwards. Then, samples were diluted, filtered with 90 mm diameter paper filters, and analysed with inductively coupled plasma atomic emission spectroscopy (ICP-AES) (Thermo Scientific, Waltham, MA, USA). Calibration curves were created with multi-element standards (SPC Science, PlasmaQual S22). Iodine and bromine were extracted, in triplicate, from 0.2 g of sample alkali digested with Tetramethylammonium hydroxide (TMAH) solution at 25% *v*/*v* in a heating graphite block system (DigiPREP MS, SCP Science, Baie-D’Urfé, QC, Canada), after being spiked with chemical standards. Then, samples were centrifuged, filtered with 0.45 µm pore size hydrophilic filters (Merck, Darmstadt, Germany) and analysed with inductively coupled plasma mass spectrometry (ICP-MS) (Thermo X series II, Thermo Fisher Scientific, Waltham, MA, USA).

### 2.4. Evaluation of Meat Quality Traits

Meat quality and carcass traits were evaluated as described in previous reports [29,30]. Briefly, carcasses were cooled for 24 h post-mortem. Then, meat pH and, after 1 h of air exposition, colour parameters (CIELAB system: lightness, L*; yellowness, b*; and redness, a*) were measured in triplicate, using a pH penetration electrode (HI9025, Hanna instruments, Woonsocket, RI, USA) and a chromameter (Minolta CR-300, Minolta camera Co. Ltd., Osaka, Japan), respectively. Afterwards, meat cooking loss and shear force were determined by cooking the breast at 80 °C (72 °C of internal temperature), followed by cooling for 2 h at room temperature. For cooking loss analysis, muscle weights were registered before and after cooking. Then, muscles strips (1 cm × 1 cm × 5 cm) were used to measure shear force with a texture analyser (TA.XTplus, Stable Microsystems, Surrey, UK) containing a Warner–Bratzler blade. The texture was expressed as a 4-replicate averaged peak value.

### 2.5. Sensory Analysis by a Trained Panel

The procedures associated with meat sensory analysis were reported by Pestana et al. [29]. Succinctly, meat was cooked at 80 °C (78 °C internal temperature) and eight samples per plaque were cut for each panellist per panel session. The ten panellists were from the trained sensory panel of the Faculty of Veterinary Medicine (University of Lisbon, Lisbon, Portugal). Tenderness, juiciness, flavour, off-flavours and overall acceptability were evaluated and classified using an 8-point scale [29], whereas flavour and off-flavour were rated from 0 (absence) to 8 (very intense).

### 2.6. Determination of Cholesterol, Vitamin E Homologs, Pigments and Minerals in Meat

Cholesterol, vitamin E homologs and pigments were determined, in duplicate, as previously described [29,30]. Briefly, cholesterol and terpenoids were extracted from 750 mg of fresh muscle samples using saponification, n-hexane extraction and HPLC analysis, according to the procedures described for *U. lactuca* and diets. Pigments were extracted from 2.5 g of fresh muscle using acetone as extraction solvent and a homogenizer for the mixture (Ultra-Turrax T25, IKA-Werke GmbH&Co. KG, Staufen, Germany). Pigment contents were measured as described for seaweed and diets, and using the formulas reported by Dere et al. [33].

The minerals were extracted from 0.3 g, or, in the case of iodine and bromine, 0.6 g of freeze-dried muscle, in accordance with procedures detailed for *U. lactuca* and diets.

### 2.7. Analysis of Meat Lipid Peroxidation

Meat lipid peroxidation was evaluated as previously reported [36]. Briefly, the concentration of thiobarbituric acid reactive substances (TBARS) was measured, in duplicate, from 1.5 g of fresh meat samples. For this purpose, a spectrophotometric method was used to detect, at 532 nm, a pink chromogen that results from the reaction between malondialdehyde (MDA) and thiobarbituric acid, and results were expressed as mg/kg of meat.

### 2.8. Evaluation of Meat Total Lipids and Fatty Acid Profile

Total lipids were extracted, in duplicate, from 0.150 g of freeze-dried meat samples using dichloromethane:methanol (2:1, *v*/*v*) solution, and gravimetrically quantified [37]. Fatty acids were methylated into FAME by basic and acidic catalysis [29] using a ratio of 19:0 methyl ester as the internal standard. Then, they were injected into a Supelcowax*^®^* 10 capillary column incorporated in a GC-FID. The standard used for FAME identification and the running conditions were described in a previous study [30]. The fatty acids were expressed as a proportion of the total fatty acids.

### 2.9. Statistics

The analysis of data was performed using the General Linear Models of SAS to perform ANOVA. Multiple comparisons of least square means adjusted with the Tukey–Kramer method (PDIFF option) were performed, considering, as a fixed factor, the treatment, and, as experimental units, the cage for ADFI and FCR or the animal for body weight, ADG and meat quality variables. Statistical power was assessed with the POWER procedure of SAS. The level of significance was considered as α = 0.05. The effect of treatments on breast chemical parameters was evaluated using principal component analysis (PCA) with SPSS Statistics for Windows (IBM Corp. released 2017, version 25.0, Armonk, NY, USA).

## 3. Results

### 3.1. Broiler´s Growth and Gastrointestinal Tract Parameters

Table 3 shows the impact of experimental diets on growth performance and gastrointestinal tract´s morphometric parameters and digesta viscosity of broilers. The final BW decreased by 13.6% in animals ingesting the ULE diet (*p* = 0.016) in comparison with the control. The average daily gain followed the same trend, with a significant reduction (*p* = 0.018) of 15.6 g/d in broilers fed the ULE diet in relation to those fed the control. For both parameters, no differences were found (*p* > 0.05) between birds fed UL or ULC diets and the control group. Moreover, FCR values did not differ (*p* = 0.172) between treatments. The total mortality observed between days 21 and 35 was 2.5% (data not shown) and included one and two broilers fed UL and ULC diets, respectively, which had severe diarrhoea. The relative weight of gastrointestinal organs was not affected by treatments, except for a decrease (*p* = 0.011) in liver weight of broiler fed macroalga containing treatments in comparison with the control. In addition, ileum length was significantly increased (*p* = 0.042) in broilers fed the ULE diet compared with those fed the control. However, only a tendency was found for larger duodenum (*p* = 0.069) and cecum (*p* = 0.060) lengths with *U. lactuca*-containing treatments. Both duodenum plus jejunum and ileum digesta viscosity were increased (*p* < 0.001) with macroalga-added diets relative to the control. However, broilers fed the ULE diet presented a lower (*p* < 0.001) ileum digesta viscosity than those fed UL and ULC diets.

### 3.2. Meat Colour and pH and Sensory Analysis

Table 4 shows the effect of dietary treatments on broilers’ carcass traits and meat sensory analysis. Meat pH, cooking loss and shear force values were not affected by treatments (*p* > 0.050). On the other hand, meat lightness (L*) and redness (a*) values were reduced (*p* < 0.001), and yellow (b*) values were increased (*p* < 0.001) with U. lactuca-containing treatments, compared with control.

The tenderness, juiciness and overall acceptability of breast meat decreased (*p* < 0.001) with macroalga treatments, relative to control. Meat flavour did not differ (*p* = 0.500) among treatments, but off-flavours were significantly increased (*p* < 0.001) by dietary seaweed.

### 3.3. Meat Lipid Peroxidation and Content of Vitamin E and Pigments

Table 5 presents the influence of dietary treatments on the oxidative stability, and diterpene and pigment profiles, of meat. Lipid peroxidation of breast was not affected (*p* > 0.050) by dietary treatments, although there was an increase (*p* < 0.050) of MDA values between day zero and day six for the control. Moreover, α- and γ-tocopherols contents decreased (*p* < 0.001) with *U. lactuca*-containing treatments, in comparison with the control. Chlorophylls a and b and β-carotene concentrations did not differ (*p* > 0.050) among treatments, but a 9-fold increase (*p* < 0.001) of total carotenoids was found with macroalga treatments, in relation to the control.

### 3.4. Meat Total Lipid Content and Fatty Acid Profile

Table 6 shows the effect of treatments on the total lipid concentration and FA profile of meat. Total lipids decreased (*p* < 0.001) with *U. lactuca*-containing treatments, compared with the control. Residual saturated fatty acids (SFA), including C12:0 and C22:0, decreased (*p* = 0.042) with ULE treatment and increased (*p* = 0.001) with UL treatment, respectively, in relation to the control. Minor monounsaturated fatty acids (MUFA), including C14:1*c*9, C16:1*c*9, and C20:1*c*11, were decreased (*p* < 0.010) by dietary macroalga, but the percentages of C18:1c11 and C22:1n-9 were higher (*p* < 0.001) with macroalga and UL treatments, respectively, than the control. However, the effect on C16:1*c*9 proportion was only significant (*p* = 0.009) for ULE treatment. Regarding PUFA, C18:3n-6 and C18:4n-3 increased (*p* < 0.001) with *U. lactuca*-containing treatments in relation to the control. Even though C22:2n-6 decreased (*p* < 0.001) with *U. lactuca*-containing treatments, C20:5n-3, C22:5n-3 and C22:6n-3 proportions were significantly higher (*p* < 0.050) with ULE treatment, than with the control, and a numerical increase in these fatty acids occurred with all macroalga treatments.

The total of SFA and *cis*-MUFA were not influenced by treatments, but the total of n-3 PUFA was significantly increased (*p =* 0.017) with ULE treatment compared with the control. The n-6/n-3 PUFA ratio was almost 2-fold lower (*p* < 0.001) with the *U. lactuca*-containing treatments than with the control.

### 3.5. Meat Mineral Profile

Table 7 presents the influence of treatments on meat mineral content. The macroalga treatments significantly increased (*p* ≤ 0.001) the most predominant minerals in meat, including magnesium, phosphorous and potassium, in relation to the control. In addition, sulphur was higher (*p* = 0.004) with the UL and ULC treatments than with the control. These results led to a 1.24-fold decrease in the sodium/potassium ratio and an increase (*p* < 0.001) of total macrominerals caused by dietary *U. lactuca*. The microminerals were also affected by the treatments, with increased bromine, copper (*p* < 0.001) and iodine (*p* = 0.002), and decreased zinc (*p* < 0.001), with macroalga treatments, in comparison with the control. Although no differences were found between treatments for the total of microminerals, the sum of macro and microminerals was significantly increased (*p* < 0.001) with *U. lactuca*-containing treatments, compared with the control.

### 3.6. Principal Component Analysis

The variables of chemical composition of breast meat for the four dietary treatments were correlated using PCA, which is shown, in two-dimensions, in Figure 1. The variability of data led to a good separation between macroalga treatments and the control, since data from UL, ULE and ULC treatments are contained in quadrants a and b, while data from the control are aggregated in quadrant d, with some points in quadrant c. The first two discriminant factors explained about 59.18% (factor 1: 33.93%, and factor 2: 25.25%) of total variability. Table 8 shows the loadings obtained with the two factors for each variable. The most discriminant variables were, for factor 1: several fatty acids including C14:0, C18:1*c*9, C18:3n-6, C20:2n-6, C20:4n-6, C22:5n-3, and C22:6n-3; and, for factor 2: C18:1*c*11, C22:2n-6, α-tocopherol, β-carotene, carotenoids, iodine and bromine.

## 4. Discussion

The incorporation of 15% *U. lactuca* in broilers´ diet to partially replace maize and soybean meal did not significantly affect chicken growth performance, although a reduction in final body weight and ADG was found with the dietary supplementation of ulvan lyase, as well as a tendency for that with the other macroalgae treatments, in comparison with broilers fed the control diet. The negative effect of the recombinant CAZyme on body weight had no repercussion on FCR, and, therefore, feed enzymes had only a residual influence on animal performance. However, these results must be interpreted with caution, since an increased sample size or the measurement of performance parameters during a higher experimental period could give different outcomes with a possible decrease in broilers´ growth with *U. lactuca*. Controversial results about the influence of feeding broilers with *Ulva* sp. on animal growth are described in the literature. Indeed, Abudabos et al. [11], Cañedo-Castro et al. [13], and Ventura et al. [10] reported no effect on ADG and FCR when feeding 3%, 6% or 10% of *Ulva* sp. to broiler chicks, but Matshogo et al. [14], Alagan et al. [12], and Nhlane et al. [15] reported an increase in FCR in broilers or hens fed up to 3–3.5% of *Ulva* sp. These differences between studies are mostly due to different alga species and growing stage of birds [1], which can modify the way animals cope with the presence of indigestible polysaccharides in algal biomass and their influence on nutrient digestibility [16]. However, few studies evaluated the effect of dietary *Ulva* sp. on chicken growth when incorporated at levels higher than 10% feed [9], which is due to the fact that 20 and 30% of seaweed were reported to reduce ADG and increase FCR in broilers [10]. In the present study, the ulvan lyase supplementation was expected to increase the bioaccessibility of nutrients, which indeed occurred for certain compounds, as, for instance, PUFA. However, this phenomenon was not enough to overcome the slight reduction in ADG in broilers fed 15% of *U. lactuca*, which might have been exacerbated by a decrease of 0.14% (as fed) of the estimated available proportion of the essential amino acid lysine in macroalga-containing diets relative to the control diet (Table 1).

Moreover, dietary *U. lactuca* had no effect on the weight and length of most of the gastrointestinal organs analysed, except for a reduction in liver weight and, when combined with ulvan lyase, an increase in ileum length. The impact of *U. lactuca* on liver was unexpected, since previous studies on broilers [14] or hens [9,15] fed either 2–3.5% [14,15] or 5–25% [9] of *Ulva* sp. reported no effects of seaweed on this organ. However, the present results are possibly caused by an increased content of algal indigestible polysaccharides, such as cellulose, in the broiler´s gut. Indeed, these compounds would decrease nutrient availability and, thus, liver metabolism with a consequent reduction on the organ size [38]. This effect of *U. lactuca* can further explain the decrease in total lipids found in the muscle of broilers fed macroalgae diets, since the liver plays a main role in lipogenesis [38]. The increase in ileum length with *U. lactuca* combined with the recombinant CAZyme might have been due to the bioactivity of algal polysaccharides, namely, ulvan, towards enterocytes proliferation and intestinal microbiota. This functional activity of polysaccharides was suggested by Cañedo-Castro et al. [13] and Guo et al. [39] after finding an increase in intestinal villus height in broilers and laying hens caused by 2 to 6% of *Ulva rigida*- and 0.5% of *Ulva* sp.-extracted polysaccharides in the diet, respectively.

The combination of *U. lactuca* and ulvan lyase reduced the ileal viscosity found with *U. lactuca* alone or in combination with the commercial CAZyme mixture, even though ulvan lyase did not completely reverse the viscosity level to that observed with the control diet. A similar effect was recently reported when broilers were fed 15% of *L. digitata* combined with a recombinant alginate lyase [28]. The present results may indicate that the ulvan lyase partially disrupted the ulvan gel-forming structure composing the *U. lactuca* cell wall, as previously observed in vitro [21], with a consequent reduction in digesta viscosity caused by the presence of seaweed. However, this activity of ulvan lyase was not reflected in an improvement of broilers´ growth performance, which suggests a minor effect of the recombinant CAZyme on feed passage and nutrient digestibility [40].

Considering meat quality traits, the breast colour was strongly influenced by dietary *U. lactuca*, with a 4-fold increase in b* value and decreases of L* and a* values. These results led to a visible yellowish meat colour for all macroalga-containing treatments. Conversely, the few studies reporting the influence of *Ulva* sp. On breast meat colour of broilers showed either no effect of 1–3% of *U. lactuca* [11] or only a time × algae dose (2–3.5% *Ulva* sp.) interaction leading to an increase in a* and L* values without any effect on b* values [14]. These differences are probably due to distinct seaweed levels in the diet, as previously reported with the microalga *Arthrospira platensis*. For instance, 15% of *A. platensis* fed to broilers increased the b* value of meat [29], but no effect was observed when feeding a lower alga dose (0.25 to 1%) [41]. To the best of our knowledge, the present study is the first to evaluate the impact on meat colour of feeding up to 15% of *Ulva* to broilers. Such high dose of macroalga promoted a significant increase on the accumulation of carotenoids in meat, which is associated with the development of yellow colour, as previously demonstrated in broilers fed up to 8% [42] and 15% [29] of *Arthrospira sp*., or in fish fed 10 or 15% of *U. rigida* [43,44]. Indeed, the zeaxanthin pigment was positively correlated with the presence of yellow colour in the breast muscle of broilers [42]. A recent study in broilers fed 15% of *L. digitata* also demonstrated an increase in carotenoids in meat caused by the presence of algae, but with a milder effect on breast colour [28]. *Ulva* sp. are generally rich in several carotenoids, including β- and α-carotenes and xanthophylls, such as astaxanthin, zeaxanthin and lutein [7,45]. These compounds have numerous health benefits acting as precursors of vitamin A and retina protectors through its accumulation in eye macula and also as antioxidant and anti-inflammatory agents [7,46]. Therefore, in the present study, the meat nutritional value of broilers was enhanced by the 8.9-fold increase in carotenoids in the breast.

Despite that, the yellow–orange colour of meat conferred by the pigments can lead to a negative perception of meat by consumers, particularly in Europe [47,48], although in some countries, such as Mexico, and, more variably and less markedly, in the United States, consumers tend to prefer a more pigmented broiler skin [49,50]. Moreover, it is worth mentioning that, at the end of the 20th century, the yellow colour of meat was generally desirable, since it was an indicator of healthier meat [47]. In current times, the search for healthy food might again stimulate the preference for a yellowish meat colour, even though no studies can confirm this aspect. In the present study, it seems that the reduction in meat acceptability with macroalga treatments was caused by a decrease in juiciness and tenderness and the presence of off-flavours (metallic, herb-like or fishy flavours) rather than by the yellowish colour of meat, since the sensory panel did not indicate the colour as a factor that compromised meat overall acceptability. Additionally, the meat was still positively scored (>4.0) for all treatments. However, a previous study [51] showed that the meat colour significantly influences the perception of consumers towards its tenderness and flavour, and thus, this aspect should not be easily disregarded. Recently, a decrease in juiciness in broiler meat was reported with 15% of dietary *A. platensis* [29] or *L. digitata* combined with a recombinant CAZyme [28]. A negative effect on overall acceptability of fillet was also observed with 10 and 15% of *U. rigida* [44], which was attributed to the change in meat colour, since there was no impact of seaweed on other sensory attributes. However, the effect of *Ulva* sp. on the sensory traits of poultry meat needs to be further explored.

The total lipids and fatty acid profile in the meat were influenced by dietary *U. lactuca*, since there was a decrease in total lipid content with the sole use of macroalga and an increase in n-3 PUFA, mostly C20:5n-3, C22:5n-3 and C22:6n-3, with *U. lactuca* combined with ulvan lyase. A similar impact of macroalga on total *n*-3 PUFA in meat was reported when broilers [52] or piglets [52] were fed 15% or 10% of *L. digitata*, respectively, although without an enzymatic effect. In spite of the present results, lean meats, with an average of 1.89% total lipids (<5%) [53], were obtained for all treatments. The increase in n-3 PUFA caused by combining *U. lactuca* with ulvan lyase indicates a degradation of alga with consequent release of nutritional and bioactive compounds. An effect of the recombinant CAZyme on releasing fatty acids from *U. lactuca* biomass was recently demonstrated in vitro, but only some MUFA, particularly C18:1*c*9, were significantly released [21]. The enrichment of poultry meat with n-3 long-chain PUFA can have various human health benefits, such as the diminishment of atherosclerosis and cardiovascular diseases [54,55]. In the present study, the 2-fold decrease in n-6/n-3 PUFA ratio in the meat from macroalga treatments relative to control could indicate a healthier meat, according to Wood*,* et al. [56], although the ratio was much higher than the recommended value (<4.0).

Furthermore, the mineral profile of meat was markedly affected by the incorporation of *U. lactuca* in broiler diet, with an increase in total macrominerals, including magnesium, phosphorus and potassium, accompanied by a decrease in sodium. Most of the microminerals were also influenced by dietary seaweed, since bromine, copper and iodine increased, and zinc decreased, with macroalga treatments.

The increased deposition of macrominerals in meat with *U. lactuca* enhances its nutritional value, considering the health benefits of magnesium, phosphorus and potassium. Indeed, magnesium is an important component of bones and is involved in oxygen uptake, energy production and electrolyte balance [57]; phosphorus participates in numerous biological processes, including ATP synthesis and bone mineralization [58]; and potassium reduces blood pressure and the risk of cardiovascular diseases [55,59]. Although caution must be taken with the high phosphorus content normally present in Western diets, the concentration of phosphorus in the meat from macroalga treatments (average of 254.7 mg/100 g) was about half of that recommended by EFSA [60] (550 mg/day) for an adult ingesting 100 g of meat per day. Moreover, the increase in potassium in meat with dietary *U. lactuca* was due to a high amount of this mineral in the alga biomass (38.822 g/kg dry matter), which is within the range described for *Ulva* sp. [1]. This occurrence explains the 1.24-fold decrease in sodium/potassium ratio in meat with the macroalga treatments. A similar result was recently observed when feeding finishing pigs with 5% of the microalga *Chlorella vulgaris* combined with a four-CAZyme mixture [61]. The reduction in sodium/potassium ratio is desirable due to the imbalance of these minerals in Western diets. Additionally, a high ratio was previously described as a good predictor of high hypertension prevalence and cardiovascular diseases [62].

The accumulation of copper and iodine in meat with the seaweed can either have positive effects on human health or cause toxicity for consumers [63,64], whereas bromine does not have recognized benefits and is considered a toxic element [65]. Although copper is a cofactor of many oxido-reductive enzymatic processes [64] and iodine plays a crucial role on the synthesis of thyroid hormone [63], pathological conditions may occur if allowances for these minerals in the diet are exceeded. Nevertheless, in the present study, feeding broilers with 15% of *U. lactuca* is not expected to have a major impact on meat nutritional quality in terms of micromineral content. Indeed, the dietary macroalga was responsible for a small increase in copper (0.068 to 0.085 mg/100 g), iodine (0.001 to 0.003 mg/100 g) and bromine (1.71 to 4.29 µg/kg body weight/day) in broiler meat leading to values much lower than the recommended daily intake for copper [64] and iodine [63] (0.90 mg and 0.15 mg /day, respectively) or the maximum allowance for bromine (1000 µg bromine/kg body weight/day) [65], for an adult person (70 kg) consuming 100 g of meat per day.

## 5. Conclusions

The inclusion of 15% *U. lactuca* in broiler chicken diet did not compromise animal growth performance and enhanced the nutritional value of meat through the accumulation of antioxidant carotenoids, n*-3* PUFA (i.e., C20:5n-3, C22:5n-3 and C22:6n-3) and macrominerals, including magnesium, potassium and phosphorus, in breast muscle. The activity of ulvan lyase promoted an increase in n-3 PUFA in meat, which were probably released from the alga biomass. However, dietary macroalga caused a modification of meat colour, from a pinkish to yellowish breast colour, and negatively influenced meat sensory analysis. Indeed, meat juiciness and tenderness decreased and off-flavours increased with alga diets, leading to a reduction in overall acceptability. In addition, ulvan lyase reduced ileal viscosity in broilers fed *U. lactuca* without a beneficial effect on animal performance, but the effect of recombinant CAZyme on the nutrient digestibility of broilers should be explored in future studies. Finally, the incorporation of lower levels (<15% feed) of seaweed in the broiler diet could represent a promising strategy to counteract the negative effect of *U. lactuca* on meat sensory perception for consumers.

## Figures and Tables

**Figure 1 animals-12-01720-f001:**
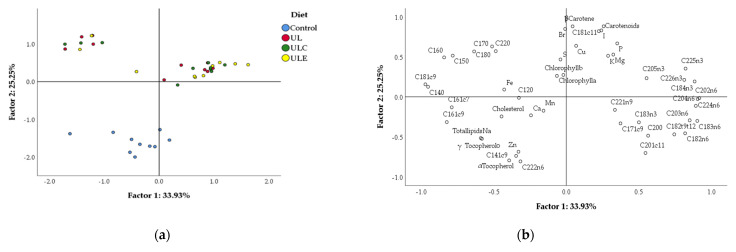
Principal component analysis of chemical composition variables of broiler breast meat: (**a**) loading plot of first and second principal factors of analysed data; (**b**) component score vectors. Experimental diets: control, maize–soybean diet; UL, control diet with 15% *U. lactuca*; ULC, control diet with 15% *U. lactuca* and 0.005% commercial enzyme mixture; ULE, control diet with 15% *U. lactuca* and 0.01% recombinant ulvan lyase.

**Table 1 animals-12-01720-t001:** Ingredients and feed supplements of dietary treatments (% as fed basis).

Item	Dietary Treatments ^1^			
	Control	UL	ULC	ULE
Maize	50.4	43.7	43.7	43.7
Soybean meal	41.2	33.2	33.2	33.2
Sunflower oil	4.80	5.98	5.98	5.98
Sodium chloride	0.38	0.00	0.00	0.00
Calcium carbonate	1.10	0.00	0.00	0.00
Dicalcium phosphate	1.60	1.40	1.40	1.40
DL-Methionine	0.120	0.170	0.170	0.170
L-Lysine	0.000	0.120	0.120	0.120
Vitamin-mineral premix ^2^	0.400	0.400	0.400	0.400
*Ulva lactuca* powder	-	15.0	15.0	15.0
Commercial enzyme mixture	-	-	0.005	-
Recombinant ulvan lyase	-	-	-	0.01

^1^ Control, maize–soybean diet; UL, control diet with 15% *U. lactuca*; ULC, control diet with 15% *U. lactuca* and 0.005% commercial enzyme mixture; ULE, control diet with 15% *U. lactuca* and 0.01% recombinant ulvan lyase. ^2^ Premix provided the following nutrients per kg of diet: vitamin A 10000 UI, pantothenic acid 10 mg, vitamin B_1_ 2 mg, vitamin B_2_ 4 mg, vitamin B_6_ 2 mg, folic acid 1 mg, cyanocobalamin 0.02 mg, vitamin D_3_ 2400 IU, vitamin K_3_ 2 mg, nicotinic acid 25 mg; vitamin E 30 mg, Cu 8 mg, Fe 50 mg, I 0.7 mg, Mn 60 mg, Se 0.18 mg, and Zn 40 mg.

**Table 2 animals-12-01720-t002:** Chemical composition of *U. lactuca* and experimental diets.

Item	Macroalgae	Dietary Treatments ^1^			
	*U. lactuca*	Control	UL	ULC	ULE
Metabolizable energy kcal/kg as DM	2664	4649	4672	4607	4597
Proximate composition (% as DM)					
Dry matter	88.7	89.0	89.2	89.3	89.4
Crude protein	28.2	23.3	23.1	22.8	23.3
Crude fat	2.85	8.8	9.0	9.8	10.3
Ash	31.7	6.6	8.8	8.8	8.9
Amino acid composition (% as fed basis)					
Arginine	-	1.54	1.26	1.26	1.26
Histidine	-	0.59	0.48	0.48	0.48
Isoleucine	-	1.12	0.91	0.91	0.91
Leucine	-	1.91	1.57	1.57	1.57
Lysine	-	1.23	1.09	1.09	1.09
Methionine	-	0.47	0.45	0.45	0.45
Phenylalanine	-	1.22	0.99	0.99	0.99
Threonine	-	0.85	0.69	0.69	0.69
Tryptophan	-	0.32	0.26	0.26	0.26
Valine	-	1.20	0.98	0.98	0.98
Fatty acid profile (% total fatty acids)					
C14:0	5.12	0.088	0.206	0.207	0.214
C16:0	22.7	9.13	8.78	8.84	8.79
C16:1*c*9	2.95	0.114	0.174	0.175	0.175
C17:0	0.454	0.051	0.050	0.049	0.047
C17:1*c*9	0.581	0.026	0.036	0.038	0.039
C18:0	1.09	3.05	3.10	3.13	3.17
C18:1*c*9	19.3	27.5	26.5	26.4	27.2
C18:2n*-6*	8.32	56.4	56.8	56.8	56.1
C18:3n-3	5.14	0.888	0.935	0.932	0.919
C18:4n-3	5.84	0.005	0.143	0.149	0.149
C20:0	0.931	0.345	0.316	0.324	0.320
Fatty acid profile (% total fatty acids)					
C20:4n-6	9.79	0.001	0.208	0.211	0.218
C20:5n-3	13.8	0.004	0.276	0.280	0.290
Diterpene profile (µg/g)					
α-Tocopherol	79.3	253.8	254.2	233.9	224.1
α-Tocotrienol ^2^	n.d	5.00	3.26	3.56	2.77
β-Tocopherol	n.d	0.723	0.767	0.736	0.723
γ-Tocopherol + β-tocotrienol	n.d	5.47	4.15	4.18	3.57
γ-Tocotrienol	n.d	6.18	4.09	4.54	3.46
δ-Tocopherol	n.d	0.961	0.689	0.729	0.701
Pigments (µg/g) ^3^					
β-Carotene	170	1.28	33.7	28.3	28.4
Chlorophyll a	2311	5.78	599	589	588
Chlorophyll b	1666	8.46	438	424	428
Total carotenoids	510	6.04	145	144	134
Mineral profile, mg/kg DM					
Arsenic	n.d.	n.d.	n.d.	n.d.	n.d.
Barium	1.26	n.d.	n.d.	n.d.	n.d.
Bromine	694	4.10	211	157	177
Cadmium	0.0493	n.d.	n.d.	n.d.	n.d.
Calcium	6202	20,263	11,315	11,195	11,756
Chromium	2.02	n.d.	n.d.	n.d.	n.d.
Cobalt	n.d.	n.d.	n.d.	n.d.	n.d.
Copper	3.73	17.8	16.4	19.1	19.7
Iodine	45.1	1.09	14.0	11.2	11.7
Iron	537	296	343	310	326
Lead	n.d.	n.d.	n.d.	n.d.	n.d.
Magnesium	25,889	2723	7826	7913	8189
Manganese	39.0	154	128	128	139
Nickel	n.d.	n.d.	n.d.	n.d.	n.d.
Phosphorous	2786	10,728	8941	8918	9284
Potassium	38,822	15,850	21,079	21,462	21,639
Sodium	52,133	3388	13,727	13,924	15,122
Sulphur	49,265	3379	14,423	14,670	15,098
Vanadium	1.25	n.d.	n.d.	n.d.	n.d.
Zinc	8.96	121	100	113	107

^1^ Control, maize–soybean diet; UL, control diet with 15% *U. lactuca*; ULC, control diet with 15% *U. lactuca* and 0.005% commercial enzyme mixture; ULE, control diet with 15% *U. lactuca* and 0.01% recombinant ulvan lyase. ^2^ Co-eluted with α-tocopherol. ^3^ Chlorophylls and carotenoids were obtained using the formulas reported by Dere et al. [33]. n.d.—not detected. DM—dry matter.

**Table 3 animals-12-01720-t003:** Growth performance for days 21 to 35, and weight, length and content viscosity of gastrointestinal tract of broilers (*n* = 10).

Item	Dietary Treatments ^1^				SEM ^2^	*p*-Value
	Control	UL	ULC	ULE		
Growth performance						
Initial body weight (g)	767	748	754	727	26.4	0.763
Final body weight (g)	1763 ^a^	1621 ^ab^	1679 ^ab^	1523 ^b^	50.9	0.016
Average daily gain (g/d)	77.4 ^a^	68.1 ^ab^	72.1 ^ab^	61.8 ^b^	3.38	0.018
Average daily feed intake (g/pen)	371	310	318	320	16.5	0.049
Feed conversion ratio	1.68	1.72	1.63	1.78	0.047	0.172
Relative weight of gastrointestinal tract (g/kg body weight)						
Crop	3.39	4.03	3.70	3.91	0.177	0.616
Gizzard	14.93	14.89	16.93	16.39	0.372	0.118
Pancreas	2.38	2.56	2.43	2.70	0.060	0.235
Liver	20.30 ^a^	17.73 ^b^	18.14 ^b^	18.2 ^b^	0.312	0.011
Duodenum	5.77	5.68	5.66	5.44	0.124	0.822
Jejunum	10.46	9.55	9.88	9.79	0.191	0.395
Ileum	8.98	9.19	9.07	8.22	0.186	0.248
Caecum ^3^	4.80	6.02	5.63	5.71	0.199	0.162
Relative length of gastrointestinal tract (cm/kg body weight)						
Duodenum	16.65	19.15	18.26	19.01	0.378	0.069
Jejunum	41.58	45.94	44.87	46.80	0.839	0.132
Ileum	46.04 ^b^	50.49 ^ab^	49.24 ^ab^	52.48 ^a^	0.835	0.042
Caecum ^3^	10.92	12.58	12.5	12.59	0.265	0.060
Content viscosity (cP)						
Duodenum + jejunum	4.75 ^b^	9.65 ^a^	10.27 ^a^	8.67 ^a^	0.466	<0.001
Ileum	6.22 ^c^	24.78 ^a^	22.01 ^a^	14.33 ^b^	1.512	<0.001

^1^ Control, maize–soybean diet; UL, control diet with 15% U. lactuca; ULC, control diet with 15% *U. lactuca* and 0.005% commercial enzyme mixture; ULE, control diet with 15% *U. lactuca* and 0.01% recombinant ulvan lyase. ^2^ SEM, standard error of the mean. ^3^ Caecum: weight of 2 caeca. ^a,b,c^ Significant differences are indicated by different superscripts within a row (*p* < 0.05).

**Table 4 animals-12-01720-t004:** Meat quality and carcass traits of broiler breast (*n* = 10).

Item	Dietary Treatments ^1^				SEM ^2^	*p*-Value
	Control	UL	ULC	ULE		
pH 24 h	5.94	5.98	5.96	5.90	0.163	0.718
Colour parameters						
Lightness (L*)	56.33 ^a^	50.63 ^b^	49.30 ^b^	50.30 ^b^	2.803	<0.001
Redness (a*)	3.77 ^a^	0.84 ^b^	1.24 ^b^	0.88 ^b^	0.755	<0.001
Yellowness (b*)	5.86 ^b^	21.69 ^a^	21.40 ^a^	21.93 ^a^	2.289	<0.001
Cooking loss (%)	31.4	28.9	28.6	31.0	1.16	0.181
Shear force (kg)	3.32	2.77	2.99	3.37	0.230	0.221
Sensorial attributes						
Juiciness	5.86 ^a^	4.98 ^b^	4.88 ^b^	4.80 ^b^	0.116	<0.001
Tenderness	6.21 ^a^	5.64 ^b^	5.55 ^b^	5.54 ^b^	0.122	<0.001
Flavour	5.75	5.41	5.36	5.42	0.116	0.500
Off-flavour	0.129 ^b^	0.622 ^a^	0.505 ^a^	0.392 ^a^	0.0920	<0.001
Overall acceptability	6.00 ^a^	5.26 ^b^	5.15 ^b^	4.97 ^b^	0.111	<0.001

^1^ Control, maize–soybean diet; UL, control diet with 15% *U. lactuca*; ULC, control diet with 15% *U. lactuca* and 0.005% commercial enzyme mixture; ULE, control diet with 15% U. lactuca and 0.01% recombinant ulvan lyase. ^2^ SEM, standard error of the mean. ^a,b^ Significant differences are indicated by different superscripts within a row (*p* < 0.05).

**Table 5 animals-12-01720-t005:** Diterpene profile and pigments in the breast meat of broilers (*n* = 10).

Item	Dietary Treatments ^1^				SEM ^2^	*p*-Value
	Control	UL	ULC	ULE		
Malondialdehyde (mg/kg)						
d 0	0.160 ^x^	0.198	0.146	0.174	0.0239	0.464
d 6	0.751 ^y^	0.510	0.439	0.456	0.1114	0.189
Diterpene profile (µg/g)						
α-Tocopherol	7.73 ^a^	4.67 ^b^	4.59 ^b^	4.49 ^b^	0.257	<0.001
γ-Tocopherol	0.0814 ^a^	0.0631 ^b^	0.0600 ^b^	0.0565 ^b^	0.00274	<0.001
Pigments (µg/100 g) ^3^						
Chlorophyll *a*	18.4	22.8	23.1	24.4	2.80	0.477
Chlorophyll *b*	33.8	40.8	40.3	43.6	4.88	0.548
β-Carotene	n.d.	0.0650	0.0550	0.0570	0.00443	0.258
Total carotenoids	54.3 ^b^	474 ^a^	517 ^a^	458 ^a^	18.34	<0.001

^1^ Control, maize–soybean diet; UL, control diet with 15% *U. lactuca*; ULC, control diet with 15% *U. lactuca* and 0.005% commercial enzyme mixture; ULE, control diet with 15% *U. lactuca* and 0.01% recombinant ulvan lyase. ^2^ SEM, standard error of the mean. ^3^ Pigments were determined using the equations described by Dere et al. [33]. ^a,b; x,y^ Significant differences are indicated by different superscripts within and between rows (*p* < 0.05). n.d.—not detected.

**Table 6 animals-12-01720-t006:** Total lipid content, cholesterol content and fatty acid composition in the breast meat of broilers (*n* = 10).

Item	Dietary Treatments ^1^				SEM ^2^	*p*-Value
	Control	UL	ULC	ULE		
Total lipids (g/100 g)	1.92 ^a^	1.47 ^b^	1.25 ^b^	1.30 ^b^	0.108	<0.001
Cholesterol (mg/g)	0.962	0.847	0.853	0.842	0.0509	0.297
FA composition (g/100 g FA)						
C12:0	0.042 ^a^	0.033 ^ab^	0.023 ^ab^	0.013 ^b^	0.0074	0.042
C14:0	0.36	0.36	0.32	0.28	0.043	0.507
C14:1*c*9	0.028 ^a^	0.000 ^b^	0.000 ^b^	0.000 ^b^	0.0027	<0.001
C15:0	0.11	0.19	0.14	0.13	0.021	0.071
C16:0	16.5	22.2	20.7	19.0	1.92	0.218
C16:1*c*7	0.45	0.41	0.39	0.33	0.033	0.074
C16:1*c*9	1.08 ^a^	0.80 ^ab^	0.75 ^ab^	0.59 ^b^	0.095	0.009
C17:0	0.22	0.31	0.23	0.28	0.025	0.039
C17:1*c*9	0.049	0.041	0.039	0.045	0.0083	0.826
C18:0	9.8	11.8	11.6	12.6	0.78	0.093
C18:1*c*9	30.4	32.3	29.6	27.8	2.35	0.598
C18:1*c*11	1.28 ^b^	1.99 ^a^	1.99 ^a^	2.11 ^a^	0.060	<0.001
C18:2n-6	33.3	22.4	25.3	26.3	3.71	0.255
C18:3n-6	0.09 ^b^	0.13 ^a^	0.14 ^a^	0.13 ^a^	0.007	<0.001
C18:2*t*9*t*12	0.22	0.16	0.18	0.18	0.039	0.645
C18:3n-3	0.51	0.40	0.42	0.38	0.042	0.194
C18:4n-3	0.03 ^b^	0.19 ^a^	0.20 ^a^	0.18 ^a^	0.017	<0.001
FA composition (g/100 g FA)						
C20:0	0.120	0.073	0.098	0.098	0.0142	0.159
C20:1*c*11	0.24 ^a^	0.11 ^b^	0.14 ^b^	0.20 ^ab^	0.024	0.002
C20:2n-6	0.458	0.461	0.583	0.697	0.0930	0.228
C20:3n-6	0.511	0.341	0.460	0.523	0.0879	0.453
C20:4n-6	2.92	2.99	3.54	4.64	0.773	0.381
C20:5n-3	0.032 ^b^	0.086 ^ab^	0.072 ^ab^	0.121 ^a^	0.0195	0.025
C22:0	0.048	0.083	0.086	0.073	0.0100	0.047
C22:1n-9	0.028 ^b^	0.076 ^a^	0.038^b^	0.011 ^b^	0.0091	0.001
C22:2n-6	0.018 ^a^	0.000 ^b^	0.000^b^	0.000 ^b^	0.0017	<0.001
C22:5n-3	0.14 ^b^	0.40 ^ab^	0.46^ab^	0.62 ^a^	0.087	0.004
C22:6n-3	0.12 ^b^	0.26 ^ab^	0.35^ab^	0.49 ^a^	0.089	0.038
Others	1.07	0.67	0.50	0.84	0.186	0.197
Σ FA (g/100 g fatty acids)						
Saturated fatty acids	26.9	35.0	33.2	32.5	2.68	0.198
Monounsaturated fatty acids	33.4	35.7	33.0	31.1	2.41	0.595
Polyunsaturated fatty acids	40.2	28.6	32.6	35.6	4.91	0.431
n-6 PUFA	39.1	27.2	31.0	33.7	4.67	0.372
n-3 PUFA	0.73 ^b^	1.27 ^ab^	1.44 ^ab^	1.75^a^	0.219	0.017
Ratios of fatty acids						
PUFA/SFA	1.55	0.99	1.18	1.24	0.201	0.321
n-6/n-3	46.5 ^a^	20.5 ^b^	20.0 ^b^	19.8 ^b^	1.75	<0.001

^1^ Control, maize–soybean diet; UL, control diet with 15% *U. lactuca*; ULC, control diet with 15% *U. lactuca* and 0.005% commercial enzyme mixture; ULE, control diet with 15% *U. lactuca* and 0.01% recombinant ulvan lyase. ^2^ SEM, standard error of the mean. Saturated fatty acids = Σ C10:0, C12:0, C14:0, C15:0, C16:0, C17:0, C18:0, C20:0, C22:0. Monounsaturated fatty acids = Σ C14:1*c*9, C16:1*c*7, C16:1*c*9, C17:1*c*9, C18:1*c*9, C18:1*c*11, C20:1*c*11, C22:1*n-*9. Polyunsaturated fatty acids = Σ C18:2*n*-6, C18:2*t*9*t*12, C18:3n-6, C18:3n-3, C18:4n-3, C20:2n-6, C20:3n-6, C20:4n-6, C20:5n-3, C22:5n-3, C22:6n-3 n-6. PUFA = Σ C18:2n-6, C18:3n-6, C20:2n-6, C20:3n-6, C20:4n-6 n-3. PUFA = Σ C18:3n-3, C18:4n-3, C20:5n-3, C22:5n-3, C22:6n-3 ^a,b^. Significant differences are indicated by different superscripts within a row (*p* < 0.05).

**Table 7 animals-12-01720-t007:** Mineral composition of breast and thigh meats of broilers (*n* = 10).

Item	Dietary Treatments ^1^				SEM ^2^	*p*-Value
	Control	UL	ULC	ULE		
Macrominerals (mg/100 g fresh weight)						
Calcium	17.9	17.5	16.6	17.8	0.40	0.140
Magnesium	31.4 ^b^	35.2 ^a^	34.8 ^a^	34.5 ^a^	0.68	0.001
Phosphorous	227 ^b^	256 ^a^	258 ^a^	250 ^a^	3.4	<0.001
Potassium	460 ^b^	522 ^a^	527 ^a^	502 ^a^	11.1	0.001
Sodium	65.8 ^a^	57.5 ^b^	56.0 ^b^	57.7 ^b^	1.52	<0.001
Sulphur	192 ^b^	207 ^a^	209 ^a^	202 ^ab^	3.4	0.004
Sodium/Potassium	0.14 ^b^	0.11 ^a^	0.11 ^a^	0.12 ^a^	0.004	<0.001
Total	994 ^b^	1095 ^a^	1102 ^a^	1064 ^a^	15.2	<0.001
Microminerals (mg/100 g fresh weight)						
Bromine	0.12 ^b^	0.30 ^a^	0.31 ^a^	0.30 ^a^	0.015	<0.001
Copper	0.068 ^c^	0.085 ^ab^	0.090 ^a^	0.079 ^b^	0.0027	<0.001
Iodine ^3^	0.001 ^b^	0.003 ^a^	0.003 ^a^	0.003 ^a^	0.0002	0.002
Iron	0.79	0.73	0.81	0.77	0.038	0.563
Manganese	0.060	0.059	0.058	0.060	0.0015	0.799
Zinc	1.15 ^a^	0.81 ^b^	0.86 ^b^	0.84 ^b^	0.040	<0.001
Total	2.19	1.96	2.13	2.06	0.074	0.200
Total macro- and microminerals	997 ^b^	1099 ^a^	1104 ^a^	1066 ^a^	15.9	<0.001

^1^ Control, maize–soybean diet; UL, control diet with 15% *U. lactuca*; ULC, control diet with 15% *U. lactuca* and 0.005% commercial enzyme mixture; ULE, control diet with 15% *U. lactuca* and 0.01% recombinant ulvan lyase. ^a,b,c^ Significant differences are indicated by different superscripts within a row (*p* < 0.05). ^2^ SEM, standard error of the mean. ^3^ The values for 8 animals from control were not detected.

**Table 8 animals-12-01720-t008:** Loadings obtained with the first two principal components of breast meat variables.

Variables	Factor 1	Factor 2
C12:0	−0.327	−0.012
C14:0	−0.953	0.126
C14:1*c*9	−0.346	−0.739
C15:0	−0.783	0.516
C16:0	−0.843	0.494
C16:1*c*7	−0.790	−0.132
C16:1*c*9	−0.824	−0.317
C17:0	−0.636	0.569
C17:1*c*9	0.374	−0.332
C18:0	−0.512	0.631
C18:1*c*9	−0.972	0.156
C18:1*c*11	0.224	0.823
C18:2n−6	0.820	−0.457
C18:3n−6	0.904	−0.301
C18:2*t*9*t*12	0.745	−0.470
C18:3n−3	0.501	−0.319
C18:4n−3	0.885	0.192
C20:0	0.564	−0.487
C20:1*c*11	0.546	−0.703
C20:2n−6	0.918	−0.016
C20:3n−6	0.851	−0.294
C20:4n−6	0.906	−0.028
C20:5n−3	0.554	0.234
C22:0	−0.486	0.572
C22:1n−9	0.335	−0.163
C222n−6	−0.316	−0.808
C224n−6	0.894	−0.113
C22:5n−3	0.823	0.354
C22:6n−3	0.814	0.212
Total lipids	−0.582	−0.525
Cholesterol	−0.447	−0.246
α−Tocopherol	−0.394	−0.796
γ−Tocopherol	−0.470	−0.615
β−Carotene	0.044	0.884
Chlorophyll *a*	−0.020	0.274
Chlorophyll *b*	−0.065	0.262
Carotenoids	0.258	0.884
Sodium	−0.590	−0.517
Potassium	0.288	0.514
Calcium	−0.243	−0.232
Magnesium	0.324	0.530
Phosphorous	0.352	0.670
Sulphur	−0.040	0.468
Copper	0.068	0.639
Zinc	−0.330	−0.687
Manganese	−0.157	−0.174
Iron	−0.428	0.091
Iodine	0.242	0.833
Bromine	−0.008	0.851

## Data Availability

The data presented in this study are available in this article.

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
