# Peer review of "Effect on Broiler Production Performance and Meat Quality of Feeding Ulva lactuca Supplemented with Carbohydrases"

_animals, 2022, doi:10.3390/ani12131720_

Round 1

Reviewer 1 Report

Animals- Costa- Macroalgae

This is a well written paper with correctly interpreted results.

I have a few minor concerns

1-The diets do not appear to be truly isonitrogenous.  There is quite a bit of difference in lysine, even with added synthetic dietary lysine.  This potentially contributed to your growth lag.

2- I know the experimental treatments were not statistically different for growth compared with controls, but they would be given a few more observations.  Be careful as you discuss that.

Ln 125 & 197 How did you chill the carcasses?

Ln 472 I am not sure I would say the US poultry market wants yellow meat.  Yellow fat- yes.

Author Response

  1. This is a well written paper with correctly interpreted results. I have a few minor concerns.

Reply: thank you for your comments; we tried to address all of them, as described below.

  1. The diets do not appear to be truly isonitrogenous. There is quite a bit of difference in lysine, even with added synthetic dietary lysine.  This potentially contributed to your growth lag.

Reply: We understand the reviewer’s concern and we agree that the difference on lysine amount could have contributed to the growth lag. This aspect is now referred in the manuscript discussion (L403 and L420 to 423, page 15).

  1. I know the experimental treatments were not statistically different for growth compared with controls, but they would be given a few more observations. Be careful as you discuss that.

Reply: The reviewer is right, as few more observations could give different results with a significant decrease of broilers´ growth when feeding 15% Ulva lactuca. This aspect is now discussed in the manuscript (L421 to 424, page 15).

  1. Ln 125 & 197 How did you chill the carcasses?

Reply: The carcasses were maintained in an air-chilled circuit (L 125, page 3).

  1. Ln 472 I am not sure I would say the US poultry market wants yellow meat. Yellow fat- yes.

Reply: The reviewer is right. The studies refer to the skin colour and acknowledge the variability that exists within the USA. However, in general, consumers from USA, especially in the Eastern regions, prefer a more pigmented (yellow) meat than those from Europe. These aspects are now more detailed between lines 474 and 482, page 16.

Reviewer 2 Report

Delete commas before “et al.” throughout manuscript. You would write “Ventura and others”, not “Ventura, and others”. Having “and others” in Latin does not affect the use : Lines 70, 161, 169, 173, 175, 211, 227, 407, 409, 433, 497

Use “algae” instead of “alga”: Lines 71, 450, 455, 463, 490, 514, 543, 546

Use “microalgae” instead of “microalga”: Lines 150, 160, Table 2, 294, 356, 363, 379, 448, 457, 473, 484, 486, 496, 516, 543

Line 70: Unbold “9” for the reference.

Line 102-103: Write out either 120 or 1 and delete comma after 120

Table 2: Might have to be broken up into multiple tables. Can it be fit on one page and still be legible?

Line 212: Line under degree symbol in parentheses – missed a track change?

Line 273: “However” seems inappropriate here since you just stated that the ileal length was longer for an algae-containing diet compared to the control and you are stating the duodenum and cecum length are also longer in algae-containing diets compared to control. I would suggest deleting “However” and adding “also” as a descriptor of the verb. Or you could reword so that it says “However, only a tendency was found…” since the previous sentence had a significant difference.

Line 288: “Table” not “Tables”.

Line 289: Delete “respectively”.

Table 6: Might have to be broken up into two tables. Can it be fit on one page and still be legible?

Figure 1: Need figures that are easier to read. Writing on the figures is blurry and hard to decipher. These look like they were screen captured from an output file, rather than inserted as their own image. I know in SAS you can export the data used to create these plots and make these plots yourself so that they are more legible. I’m not as familiar with SPSS though.

Table 8: Can it be fit on one page and still be legible?

Line 429: Micro vs. macro? You’ve been using micro earlier in the manuscript. Double-check this.

Lines 470-472: Do you think that the sensory results would be different in the US or Mexico since the color results are interpreted differently? If consumers in the US and Mexico are already used to eating yellow breasts, then they might not associate the flavor of feeding the algae as an off-flavor. Just something to think about.

Line 484: “Sole” instead of “solely”.

Author Response

  1. Delete commas before “et al.” throughout manuscript. You would write “Ventura and others”, not “Ventura, and others”. Having “and others” in Latin does not affect the use : Lines 70, 161, 169, 173, 175, 211, 227, 407, 409, 433, 497

Reply: Thank you for your correction. The commas before “et al.” were deleted throughout manuscript

  1. Use “algae” instead of “alga”: Lines 71, 450, 455, 463, 490, 514, 543, 546

Reply: “Alga” was replaced by “algae” in lines 454, 459, 467, 505, 529, 558, 562.

  1. Use “microalgae” instead of “microalga”: Lines 150, 160, Table 2, 294, 356, 363, 379, 448, 457, 473, 484, 486, 496, 516, 543

Reply: “Macroalga” was replaced by “macroalgae” in lines 150, 160, Table 2, 295, 357, 364, 380, 453, 462, 484, 500, 502, 512, 532, 559.

  1. Line 70: Unbold “9” for the reference.

Reply: Done.

  1. Line 102-103: Write out either 120 or 1 and delete comma after 120

Reply: We maintained both 120 and 1 to indicate the number of animals and their age, respectively. However, we spelled one hundred and twenty instead of 120 and deleted the comma afterwards.

  1. Table 2: Might have to be broken up into multiple tables. Can it be fit on one page and still be legible?

Reply: Table 2 was broken up into two Tables.

  1. Line 212: Line under degree symbol in parentheses – missed a track change?

Reply: It is now corrected (L 218, page 7).

  1. Line 273: “However” seems inappropriate here since you just stated that the ileal length was longer for an algae-containing diet compared to the control and you are stating the duodenum and cecum length are also longer in algae-containing diets compared to control. I would suggest deleting “However” and adding “also” as a descriptor of the verb. Or you could reword so that it says “However, only a tendency was found…” since the previous sentence had a significant difference.

Reply: The sentence was rewritten as suggested “However, only a tendency was found…” (L 279 to 280, page 8).

  1. Line 288: “Table” not “Tables”.

Reply: Done.

  1. Line 289: Delete “respectively”.

Reply: Done.

  1. Table 6: Might have to be broken up into two tables. Can it be fit on one page and still be legible?

Reply: Table 6 was broken up into two Tables.

  1. Figure 1: Need figures that are easier to read. Writing on the figures is blurry and hard to decipher. This look like they were screen captured from an output file, rather than inserted as their own image. I know in SAS you can export the data used to create these plots and make these plots yourself so that they are more legible. I’m not as familiar with SPSS though.

Reply: Thank you for your suggestion. Figure 1 (a, b) was optimized by exporting the image from SPSS output and converting it into a 600 dpi image using the XL Toolbox NG application for Excel.

  1. Table 8: Can it be fit on one page and still be legible?

Reply: Table 8 was broken up into two Tables, as well as all the other Tables that could not fit in one page.

  1. Line 429: Micro vs. macro? You’ve been using micro earlier in the manuscript. Double-check this.

Reply: “Microalgae” or “microalga” was only used in lines 82, 473 and 548 to refer either the activity of CAZymes (L 82) or similar results found with microalgae (L 473 and 548). For the rest of the manuscript, the term “macroalgae” or “macroalga” was used (double-checked).

  1. Lines 470-472: Do you think that the sensory results would be different in the US or Mexico since the colour results are interpreted differently? If consumers in the US and Mexico are already used to eating yellow breasts, then they might not associate the flavour of feeding the algae as an off-flavour. Just something to think about.

Reply: The reviewer raised a good point of view. These aspects are now more detailed between lines 490 and 508, page 17. Indeed, a different colour perception can influence the sensory analysis of consumers, as reported by Kennedy et al. (2005). However, the consumers´ perception of colour is variable, particularly in USA. For instance, in the Eastern United States, deeply pigmented birds are desirable, whereas in the Northwestern USA, pale skin colour is preferred (Fletcher, 1999). Moreover, in our opinion, the fact that some consumers in the USA and Mexico are used to yellow meat does not make them familiar with the algae flavour, because the commercial-type pigments responsible for increasing meat pigmentation are mostly synthetic apo-ester and canthaxanthin or natural xanthophylls derived from marigold extract (Tagetes erecta), corn or alfalfa, and only occasionally from algae (Castañeda et al. 2005).

  1. Line 484: “Sole” instead of “solely”.

Reply: “Solely” was replaced by “Sole” in line 515, page 15.